# Photoprotective Effects of Processed Ginseng Leaf Administration against UVB-Induced Skin Damage in Hairless Mice

**DOI:** 10.3390/molecules28186734

**Published:** 2023-09-21

**Authors:** Eunjung Son, Yun Mi Lee, Seung-Hyung Kim, Dong-Seon Kim

**Affiliations:** 1KM Science Research Division, Korea Institute of Oriental Medicine, Daejeon 34054, Republic of Korea; ejson@kiom.re.kr (E.S.); candykong@kiom.re.kr (Y.M.L.); 2Institute of Traditional Medicine and Bioscience, Daejeon University, Daejeon 34520, Republic of Korea; sksh518@dju.kr

**Keywords:** *Panax ginseng* C.A. Mayer, ginseng leaf, acid reaction, skin photoaging, ultraviolet irradiation

## Abstract

Although ginseng leaves contain a larger amount of ginsenosides than the roots, studies on the protective effect of oral administration of ginseng leaves against photoaging are lacking. Processed ginseng leaves (PGL) prepared by acid reaction to increase effective ginsenoside content showed higher levels of Rg3 (29.35 mg/g) and Rk1 (35.16 mg/g) than ginseng leaves (Rg3 (2.14 mg/g) and Rk1 (ND)), and ginsenosides Rg3 and Rk1 were evaluated as active ingredients that protected human keratinocytes against UVB-induced cell damage by increasing cell proliferation and decreasing matrix metalloproteinase (MMP)-2 and 9 secretion. Herein, the effect of oral PGL administration (50, 100, or 200 mg/kg, daily) against photoaging in HR-1 hairless mice was assessed by measuring wrinkle depth, epidermal thickness, and trans-epidermal water loss for 16 weeks. The PGL treatment group showed reduced skin wrinkles, inhibited MMP-2 and MMP-9 expression, and decreased IL-6 and cyclooxygenase-2 levels. These data suggest that oral PGL administration inhibits photoaging by inhibiting the expression of MMPs, which degrade collagen, and inhibiting cytokines, which induce inflammatory responses. These results reveal that ginseng leaves processed by acid reaction may serve as potential functional materials with anti-photoaging activities.

## 1. Introduction

The skin is directly exposed to air pollution, tobacco smoke, sunlight, and other environmental factors [1,2,3]. Ultraviolet (UV) radiation is a prominent external factor contributing to skin aging, commonly referred to as photoaging. This phenomenon occurs concurrently with the normal aging process [2,4]. Photoaged skin is characterized by roughness, pronounced wrinkles, and pigmentation [2,5,6].

Chronically sun-exposed skin is characterized by increased wrinkle formation, which is thought to be the result of equilibrium disturbance of the accumulation and degradation of extracellular matrix proteins, such as collagen [7]. The degradation of collagen is typically tightly controlled by the activity of matrix metalloproteinases (MMPs) and their natural inhibitors. MMP-1 produced by dermal fibroblasts and epidermal keratinocytes cleaves type I collagen into specific fragments. These fragments are then further hydrolyzed by MMP-2 and MMP-9, thereby inhibiting wrinkle production [8].

Currently, synthetic and mineral UV filters are widely recommended for the protection of skin against UV. However, there has been a growing debate regarding the safety of these molecules owing to their association with comedones, contact dermatitis, photosensitivity, and endocrine disorders [9,10]. Therefore, extensive research has been conducted to examine the anti-inflammatory and antioxidant properties of naturally occurring botanical compounds, such as flavonoids, polyphenols, and monoterpenes [5,11]. Modern approaches to the utilization of such herbal compounds include topical formulations and dietary supplementation [12,13,14].

Oral photoprotectants do not directly protect the skin against damage caused by high-energy photons. Therefore, they have no effect on erythema or other harmful effects of sun exposure. However, their user-friendly nature renders them advantageous in multiple ways. In addition, their efficiency is not affected by external conditions, and their half-life can be determined through pharmacological methods. Furthermore, their effectiveness remains consistent, regardless of the degree of absorption through the skin. The ideal photoprotectant is a skin-friendly oral photoprotectant [15,16,17].

Ginseng (*Panax ginseng* Meyer) is a perennial plant belonging to the family *Araliaceae* and has been used as a medicinal plant or as a natural tonic in many Asian countries for more than 2000 years. Ginsenosides are responsible for the majority of pharmacological efficacies of ginseng [18]. They are categorized into three types based on aglycone moieties: protopanaxadiol (PPD)-type ginsenosides (Ra1, Ra2, Ra3, Rb1, Rb2, Rb3, Rc, Rd, Rg3, Rh2, etc.), protopanaxatriol (PPT)-type ginsenosides (Re, Rg1, Rg2, Rf, Rh1, etc.), and oleanolic acid-type ginsenosides (Ro, etc). Many studies have shown that oral ingestion of ginseng roots yields notable beneficial effects in humans [19]. However, the pharmacological advantages of ginseng leaves have not been comprehensively investigated. In addition to their cost-effectiveness, ginseng leaves contain many types of bioactive ingredients, such as polysaccharides, phytosterols, flavonoids, polyacetylene alcohols, peptides, and a greater amount of ginsenosides Rb1, Rb2, Rd, Rc, and Re than ginseng roots [20,21,22]. Deglycosylated ginsenosides (Rg3, Rh2, etc.) are more easily absorbed into the body and exhibit relatively high levels of pharmacological activity compared to major ginsenosides [23,24]. The deglycosylation of ginsenosides can be achieved using liquid-based catalysts, solid acid catalysts, and fermentation methods using food-grade enzymes, food-compatible microorganisms, and immobilized enzymes [25,26]. Nevertheless, enzymes are generally expensive, and acid hydrolysis may be corrosive, unstable, and non-specific.

Research has shown that wrinkles can be improved by topical application of ginseng root and its processed products [27]. However, there is currently no research on wrinkle improvement through oral administration of standardized ginseng leaves with increased amounts of minor ginsenosides through acid hydrolysis and ginseng leaf extract. In this study, we tried to confirm that oral administration of processed ginseng leaf (PGL) with increased minor ginsenoside content had a significant effect on UV-induced skin wrinkles in hairless mice.

## 2. Results and Discussion

### 2.1. Examination of PGL and Ginsenoside Content

As shown in Table 1, a comparison of the HPLC chromatograms of unprocessed and PGL extracts illustrated that this process transformed primary ginsenosides, such as Rg1, Re, Rb1, Rc, Rb2, and Rd, into deglycosylation and/or dehydrated ginsenosides, such as Rg3 and Rk1. The content of ginsenoside Rg3 in PGL increased by more than 10 times that of ginseng leaf extract, and Rk1 was also newly produced (Appendix A). The PGL produced by the acid reaction yielded higher amounts of ginsenosides Rg3 and Rk1 than ginseng, red ginseng, and ginseng leaves [28,29,30]. Ginseng leaves are a valuable source of ginsenosides. Moreover, hydrolysis is a useful processing method that increases the content of Rg3 and Rk1 by 10-fold.

### 2.2. Protective Effects of PGL against UVB-Induced Damage in HaCaT Cells

Before analyzing the protective effects of the ginsenosides Rg3 and Rk1 on UVB-induced damage in HaCaT cells, we evaluated the cytotoxicity of ginsenosides Rg3 and Rk1 at various concentrations (5, 10, 20, and 40 μg/mL). The results showed that the ginsenosides Rg3 and Rk1 were not toxic at any concentration and that they promoted cell proliferation at concentrations higher than 10 μg/mL (Figure 1A). The viability of HaCaT cells was evaluated following UVB irradiation. We investigated the cytoprotective effects of ginsenosides Rg3 and Rk1 at a concentration of 40 mJ/cm^2^, which causes cell damage. UVB-induced damaged HaCaT cells were treated with the ginsenosides Rg3 and Rk1. Consequently, the cell viability of untreated cells was reduced to 65.9% compared to that of the control group. However, 20 and 40 μg/mL ginsenosides Rg3 and Rk1 led to significant amelioration of UVB-induced damage (Figure 1B).

UV irradiation induces the expression of MMPs, such as MMP-2 and MMP-9, resulting in the degradation of various extracellular matrix components and characteristic changes associated with skin photoaging [31]. The expression of MMP-2 and MMP-9 was evaluated using a Quantikine enzyme-linked immunosorbent assay (ELISA) Kit (Molecular Devices, LLC, San Jose, CA, USA). When HaCat cells were exposed to UVB irradiation, the secretion of MMP-2 and MMP-9 increased. With increasing concentrations of the ginsenosides Rg3 and Rk1, the secretion of MMP-2 and MMP-9 was inhibited compared to that in the UVB-irradiated cells. The concentrations of MMP-2 and MMP-9 were 5-fold higher in the UVB irradiation group than in the control group. In contrast, the levels of MMP-2 (Figure 1C) and MMP-9 (Figure 1D) were significantly reduced in the ginsenosides Rg3 and Rk1 treatment groups, respectively (*p* < 0.005). Thus, these results suggest that the ginsenosides Rg3 and Rk1 inhibit MMP expression and wrinkle formation due to aging.

### 2.3. Body Weight Change

To investigate the effects of 50, 100, and 200 mg/kg PGL treatments and UVB irradiation on the body weight of HR-1 mice, we measured and recorded body weight at weeks 8 and 16 during the treatment process. As shown in Figure 2, the vehicle group exhibited slightly reduced body weight, but there was no significant difference. However, in the normal and PGL administration groups, body weight increased significantly at week 16 (*p* < 0.005). It was confirmed that the vehicle group had a significant difference in body weight compared with that in the normal group at week 16 (*p* < 0.01). Exposure to artificial UVB causes stress to animal skin and weight loss, validating that UVB served as a stress-inducing agent in mice, but PGL administration was effective in maintaining body weight.

### 2.4. Effect of PGL on Wrinkle Formation Induced by UV Irradiation

UV, the main cause of DNA damage, has been reported to induce MMPs in skin tissue, which can break down collagen proteins and the skin layer, resulting in the loss of elasticity and wrinkle formation [32].

After 12 weeks of UVB irradiation, the depth of wrinkles in the vehicle group increased compared to that in the normal group. However, in the PGL-treated group, the depth of the wrinkles was lower than that in the vehicle control group. After 16 weeks of UVB exposure, we validated that the mice in the vehicle group had developed many wrinkles on the dorsal skin due to UV irradiation (Figure 3A). Moreover, we validated that PGL administration decreased wrinkle formation in a dose-dependent manner (Figure 3B). These results suggest that MMP expression induced by UV irradiation can be restored to normal levels by oral administration of PGE.

### 2.5. Effect of PGL on Epidermal Thickness of the Dorsal Skin of Mice Exposed to UV

UV radiation can increase the thickness of the epidermis and cause the skin to become thicker. UV-irradiated mice (vehicle) exhibited thicker epidermis than mice in the control group, which were not exposed to UV irradiation. In the PGL-treated groups, epidermal thickness was significantly reduced in a dose-dependent manner (*p* < 0.01) compared to that in the vehicle group (Figure 4A). As shown in Figure 4B, hematoxylin and eosin (H&E) staining revealed that PGL administration significantly decreased epidermal thickness. Moreover, we observed histological changes in the dorsal skin.

### 2.6. Effect of PGL on Transepidermal Water Loss (TEWL) in the Dorsal Skin of Mice Exposed to UV

To evaluate the protective effect of PGL against water loss, we analyzed the UVB-irradiated dorsal skin of mice (Figure 5). UVB irradiation led to increased TEWL of skin by 4.2-fold compared to that in the normal group. However, PGL administration decreased TEWL compared to the vehicle group. These results indicate that PGL is safe and effective in preventing photoaging.

### 2.7. Effect of PGL on UVB-Induced Expression of Wrinkle-Related Genes

MMP expression is increased by UV irradiation [33]. Accordingly, the mRNA expression of MMP-2 and -9 was examined to determine whether PGE oral administration can reduce MMP expression after UVB irradiation (Figure 6). The expression of MMP-2 and MMP-9 was significantly upregulated in the vehicle group compared to that in the normal group (*p* < 0.01). However, PGL administration (100 and 200 mg/kg) significantly decreased the expression levels of these genes (*p* < 0.05). These results suggest that MMP expression induced by UV irradiation can be restored to normal levels by oral administration of PGE.

### 2.8. Effect of PGL on UVB-Induced Inflammation

COX-2 is an inflammatory mediator, while IL-6 is a proinflammatory cytokine [34]. Such inflammatory mediators further induce collagen degradation by promoting apoptosis in dermal fibroblasts, enhancing the expression of MMP-2 and MMP-9, and inhibiting the expression of procollagen [35]. The production of the pro-inflammatory cytokines IL-6 and COX-2 in the skin tissue of hairless mice was examined by real-time polymerase chain reaction (RT-PCR) (Figure 7). UVB exposure increased IL-6 and COX-2 expression. PGL administration (100 and 200 mg/kg) significantly attenuated the expression of IL-6 and resulted in a decline of COX-2 expression (*p* < 0.05). Thus, PGL administration can reduce skin wrinkles caused by inflammation. These results signify that PGL administration ameliorated UVB-induced skin inflammation.

## 3. Materials and Methods

### 3.1. Materials

Ethanol, acetonitrile, methanol, and water were of HPLC grade (J. T. Baker, Phillipsburg, NJ, USA). Acetic acid was purchased from Sigma-Aldrich (St. Louis, MO, USA). The reference standards, ginsenosides Rg1, Re, Rg2, Rb1, Rc, Rb2, Rb3, Rd, Rg3, and Rk1, were purchased from Chemfaces (Wuhan, Hubei, China). Dulbecco’s modified Eagle’s medium (DMEM), penicillin/streptomycin, and fetal bovine serum (FBS) were purchased from Gibco BRL (Grand Island, NY, USA). Human total MMP-2 and MMP-9 ELISA kits were obtained from R&D Systems (Minneapolis, MN, USA).

### 3.2. Preparation of Processed Ginseng Leaf Extract (PGL)

Dried ginseng leaves were purchased from Shaanxi EDW Biotech Co., Ltd. (Xian, China). The leaf tissue (500 g) was extracted with 50% ethanol for 4 h in vacuo, evaporated, and then freeze dried to obtain 618.5 g of ginseng leaf extract (yield 23.7%). Ginseng leaf powder (100 g) was rehydrated using 400 mL of water, and 12 mL of acetic acid was added. The mixture was refluxed for 1 h, and a precipitate was obtained. The precipitate obtained was then washed with water, freeze-dried, and labeled as PGL.

### 3.3. Analytical Conditions

An Ultra Performance Liquid Chromatography (UPLC, Waters, MA, USA) System equipped with a quaternary pump, auto-sampler, and photodiode array detector with Acquity UPLC^®^ BEH C18, 100 ×  2.1 mm, 1.7 μm was used for analysis. Elution was performed using solvent A (water) and solvent B (acetonitrile) in a gradient elution at a flow rate of 0.5 mL/min as follows: 0–5 min, 18–18% B; 5–7 min, 18–20% B; 7–12 min, 20–28% B; 12–18 min, 28–30% B; 18–25 min, 30–35% B; 25–28 min, 35–35% B; 28–30 min, 35–40% B; 30–38 min, 40–40% B; 38–40 min, 40–46% B; 40–45 min, 46–46% B; 45–47 min, 46–100% B; 47–48 min, 100–18% B; 48–50 min, and 18–18% B. The detection wavelength was 200 nm. The column temperature was maintained at 40 °C, and the injection volume was 2 µL.

### 3.4. Cell Culture and UVB Irradiation

Human dermal keratinocytes (HaCaT, Korean Cell Line Bank, Seoul, Republic of Korea) were cultured in DMEM containing 10% (*v*/*v*) FBS and antibiotics for 24 h in a 37 °C, 5% CO_2_ incubator. Cells were sub-cultured every 2–3 days, and 120 mJ/cm^2^ of area was irradiated with UV rays (wavelength, 31 nm) to induce cell damage. UVB irradiation and cell treatment were performed as follows: cells were treated with various concentrations of ginsenosides Rg3 and Rk1 (5, 10, 20, 40, and 80 μg/mL) for 24 h and then exposed to UVB irradiation at a dose of 40 mJ/cm^2^ for 1 min (UV-X000; LAB24, Seoul, Republic of Korea). The final UVB irradiation intensity on the upper surface of the plate was 0.6 mW/m^2^. Cells not pre-treated or exposed to UVB irradiation were used as a control group.

### 3.5. Cytotoxicity Assay

HaCaT cells were seeded in 100 μL of medium in a 96-well plate (1 × 10^4^ cells/well). After 24 h of incubation, cells were treated with ginsenosides Rg3 and Rk1 at the indicated concentrations and subjected to UVB irradiation for an additional 24 h. The effects of Rg3 and Rk1 on keratinocytes were determined by MTT (3-(4,5-Dimethylthiazol-2-yl) assay (Sigma-Aldrich Chemical Co., St. Louis, MO, USA). MTT solution (50 µg) was added to each well, and incubation was performed at 37 °C for 4 h. The supernatant was discarded, and the formazan crystals were dissolved in 100 µL of dimethyl sulfoxide. Cell viability was evaluated by measuring absorbance using a microplate reader (Bio-Rad, Hercules, CA, USA).

### 3.6. Evaluation of MMP-2 and MMP-9 Secretion

HaCaT cells were seeded in 96-well plates (5 × 10^4^ cells/well) and pretreated with the ginsenosides Rg2 and Rk1 using the same protocol. Cells were exposed to UVB radiation, and the supernatant of culture medium was obtained and centrifuged at 189× *g* for 10 min. The concentration of MMP-2 and MMP-9 in the culture medium was determined using a microplate reader (Molecular Devices, LLC, San Jose, CA, USA) using MMP-2 (Cat. No. DMP200) and MMP-9 (Cat. No. DMP900) ELISA kits.

### 3.7. Experimental Animals

Hos:HR-1 hairless mice (6 weeks old) were purchased from Orient Bio (Seongnam, Republic of Korea). Mice were individually housed under conditions of a 12-h light and dark cycle and 22 ± 2 °C with a relative humidity of 50 ± 10%. The mice had ad libitum access to food and water. All animal experimental protocols were reviewed and approved by the Animal Protection Committee of Daejeon University (Daejeon, Republic of Korea; DJUARB2022-029).

### 3.8. PGL Treatment and Experimental Design

HR-1 hairless mice were randomly divided into five groups (n = 5 per group): (1) normal untreated controls (normal); (2) untreated UV-induced vehicle (vehicle); (3) UV-induced + 50 mg/kg PGL (PGL 50); (4) UV-induced + 100 mg/kg PGL (PGL 100); and (5) UV-induced + 200 mg/kg PGL (PGL 200). Mice in the PGL treatment group received oral administration of PGL by gavage daily.

### 3.9. UV Irradiation and Body Weight

The dorsal skin of HR-1 mice was subjected to UVB irradiation using a UVB lamp (15 W; UV intensity, 100 μW/cm^2^; maximum wavelength, 312 nm; Ieda Boeki Co., Tokyo, Japan). For 12 weeks, dietary intake and body weight were measured at regular intervals each week [36,37]. Body weight was measured and recorded on Mondays on weeks 8 and 16 to investigate the effects of PGL treatment and UVB irradiation on the body weight of HR-1 mice.

### 3.10. Skin Wrinkles and TEWL

The degree of skin wrinkles caused by UVB was determined by observing the formation of wrinkles. To evaluate the formed wrinkles, we anesthetized HR-1 mice in the 16th week by injecting them intraperitoneally with chloral hydrate. Following exposure to UVB irradiation exposure, skin wrinkles were measured at weeks 10, 12, 14, and 16 using Double-stick Disc (3M, Neuss, Germany) and DETAX system II (MIXPAC, Sulzer Ltd., Winterthur, Switzerland). The Double-Stick Disc was stuck to the mouse skin and removed 2–3 min later. The wrinkles formed on the disc were evaluated based on the scoring system reported by Tsukahara [36]. According to the scoring system, wrinkle-free skin is level 0, multiple shallow wrinkles is level 1, multiple wrinkles is level 2, and multiple deep wrinkles is level 3. After removing the disc, we washed the skin with 70% ethanol to visually analyze and photograph skin wrinkles using a USB Digital Microscope (×400; CE FOROHS, Shenzhen, Guangdong, China). TEWL was measured on week 16 using the TM 300 Tewameter (Courage & Khazaka Electronics, Cologne, Germany).

### 3.11. RT-PCR Analysis

Mice were euthanized after 16 weeks of UVB irradiation, and total RNA was extracted from the dorsal skin tissue using the RNeasy Mini Kit (Qiagen, Hilden, Germany). Total RNA was converted to cDNA using the iScript cDNA Synthesis Kit (Bio-Rad) according to the manufacturer’s protocol. After cDNA synthesis, quantitative RT-PCR was conducted using an ABI StepOnePlus™ Real-Time PCR System (Applied Biosystems, Foster City, CA, USA) and iQ SYBR Green Supermix (Bio-Rad). The sequences of the primers used in this study are shown in Table 2. The PCR conditions were denaturation at 95 °C for 30 s, annealing at 95 °C for 15 s, and extension at 72 °C for 60 s. The data were evaluated using the ∆∆Ct method and expressed relative to GAPDH.

### 3.12. Histological Observation of Skin

After 16 weeks, the dorsal skin of the mice located between the ilia was harvested under anesthesia, and later, the mice were euthanized through a sodium pentobarbital overdose. The skin was fixed with 4% paraformaldehyde for 24 h, and then a 30 µm frozen section was obtained. Histological characteristics and epidermal thickness were investigated by H&E staining. Histological changes were investigated and imaged using a light microscope (Olympus, Tokyo, Japan).

### 3.13. Statistical Analyses

All data are presented as the mean ± standard error of the mean. Significant differences were analyzed using a one-way analysis and Dunnett’s test. All analyses were performed using GraphPad Prism 7.0 (GraphPad Software, San Diego, CA, USA). Statistical significance was set to a *p*-value of less than 0.05.

## 4. Conclusions

Compared to ginseng roots, ginseng leaves contained a greater amount of beneficial ginsenosides. PGL exerted an inhibitory effect on UV-induced skin aging. This effect was achieved by increasing the content of minor ginsenosides, such as Rg3 and Rk1, in ginseng leaves through acid reactions. Following PGL administration for 16 weeks in HR-1 hairless nude mice, the mean depth of skin wrinkles, epidermal thickness, TEWL, MMP-2, MMP-9, and IL-6 improved in a dose-dependent manner. Therefore, oral administration of PGL may be an effective herbal remedy for skin aging caused by UV damage and should be verified in future clinical studies.

## Figures and Tables

**Figure 1 molecules-28-06734-f001:**
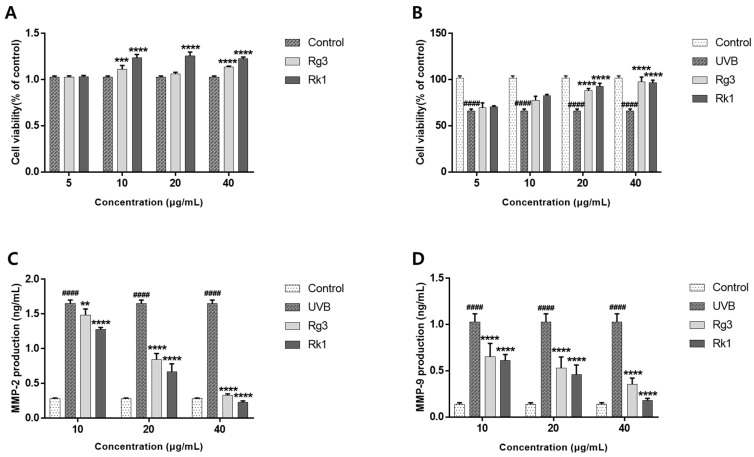
Cell viability of human keratinocytes after ultraviolet B (UVB) exposure and the effects of the ginsenosides Rg3 and Rk1 on matrix metalloproteinase (MMP)-2 and MMP-9 secretion by HaCaT cells exposed to UVB. Cells were treated with the ginsenosides Rg3 and Rk1 before UVB irradiation. HaCaT cell viability after pretreatment with the ginsenosides Rg3 and Rk1 at various concentrations (**A**). Cell viability of cells treated with the ginsenosides Rg3 and Rk1, followed by UVB irradiation (**B**). The levels of secreted MMP-2 (**C**) and MMP-9 (**D**) were measured in the culture medium of UVB-irradiated HaCaT cells. ^####^
*p* < 0.001 vs. control group. ** *p* < 0.01 vs. control group. *** *p* < 0.005 vs. control group. **** *p* < 0.001 vs. control group.

**Figure 2 molecules-28-06734-f002:**
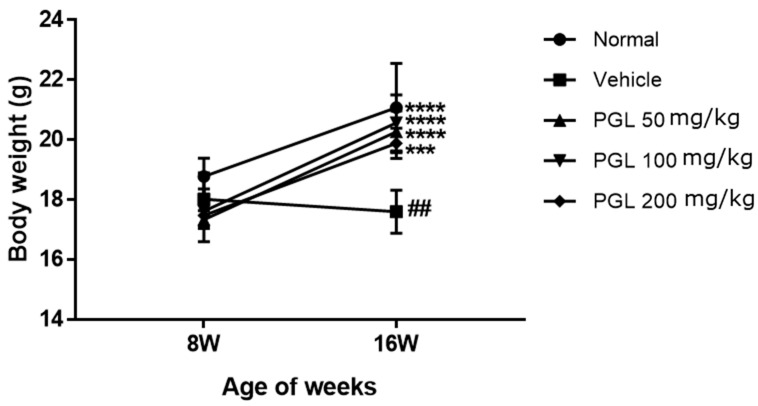
Body weight changes in HR-1 hairless mice exposed to UVB irradiation. ^##^
*p* < 0.01 vs. normal group. *** *p* < 0.005 vs. 8W test group. **** *p* < 0.001 vs. 8W test group (Figure 1B).

**Figure 3 molecules-28-06734-f003:**
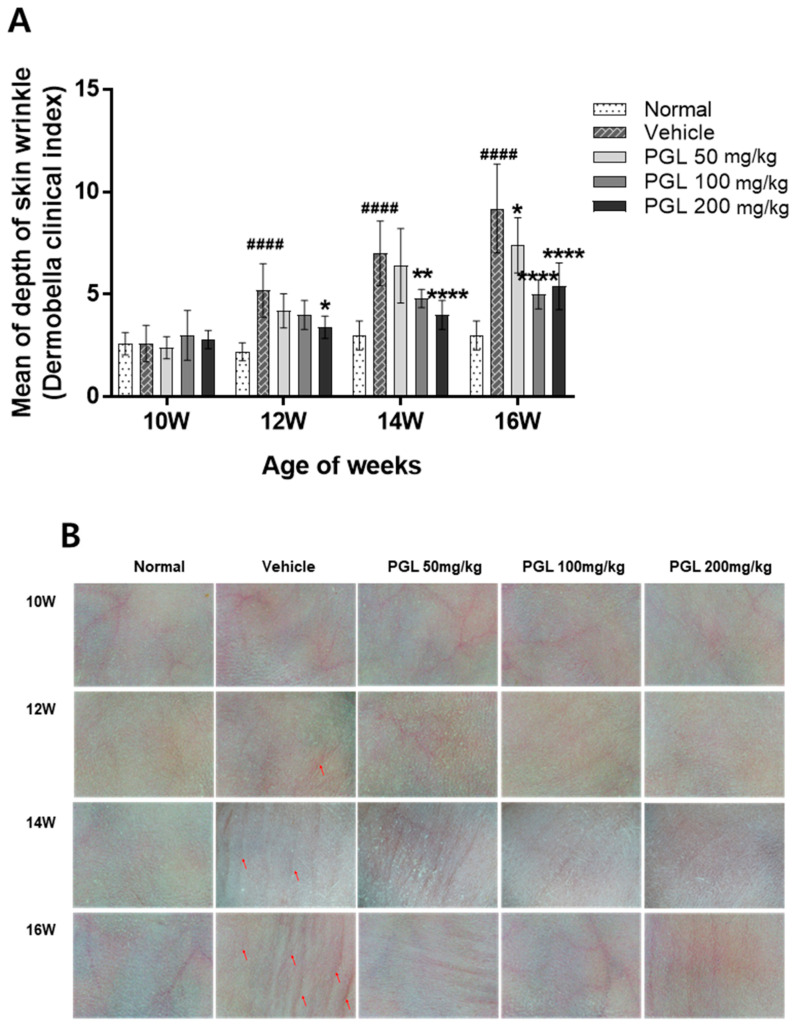
Effects of PGL on the mean depth of UVB-induced skin wrinkles (**A**). The analysis of skin dermobella obtained from the dorsal skin of hairless mice exposed to UVB irradiation (**B**). Values are mean ± S.E.M. for five mice. ^####^
*p* < 0.001 vs. normal group. * *p* < 0.05 vs. vehicle group. ** *p* < 0.01 vs. vehicle group. **** *p* < 0.001 vs. vehicle group.

**Figure 4 molecules-28-06734-f004:**
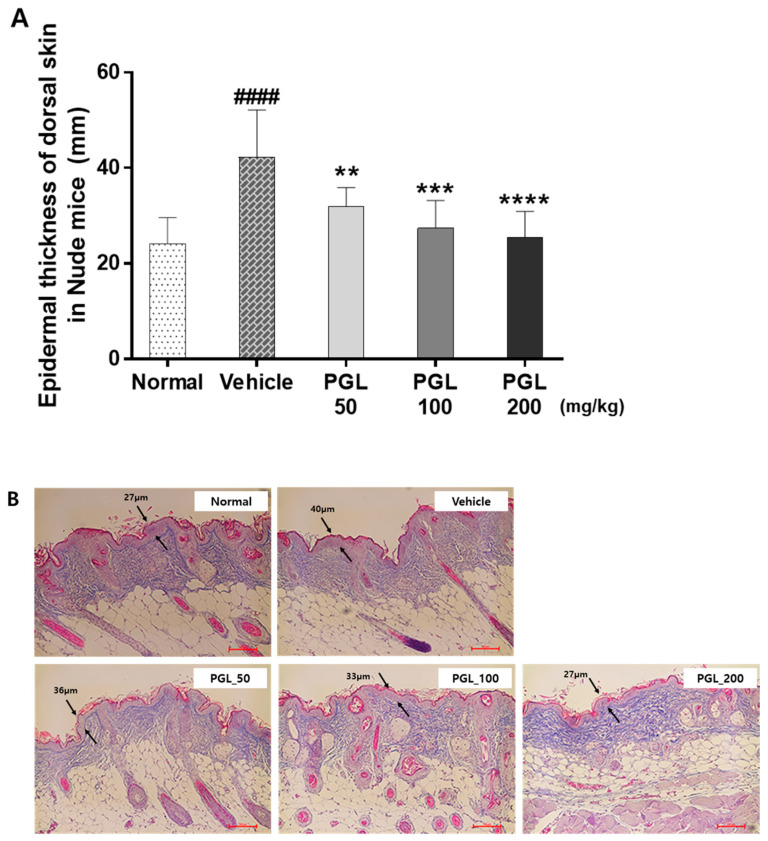
Epidermal thickness of the dorsal skin of mice exposed to ultraviolet irradiation (**A**). Analysis of epidermal thickness by hematoxylin and eosin staining. Scale bar = 100 µm (**B**). Values are expressed as the mean ± S.E.M. for five mice. ^####^
*p* < 0.001 vs. normal group. ** *p* < 0.01 vs. vehicle group. *** *p* < 0.005 vs. control group. **** *p* < 0.001 vs. vehicle group.

**Figure 5 molecules-28-06734-f005:**
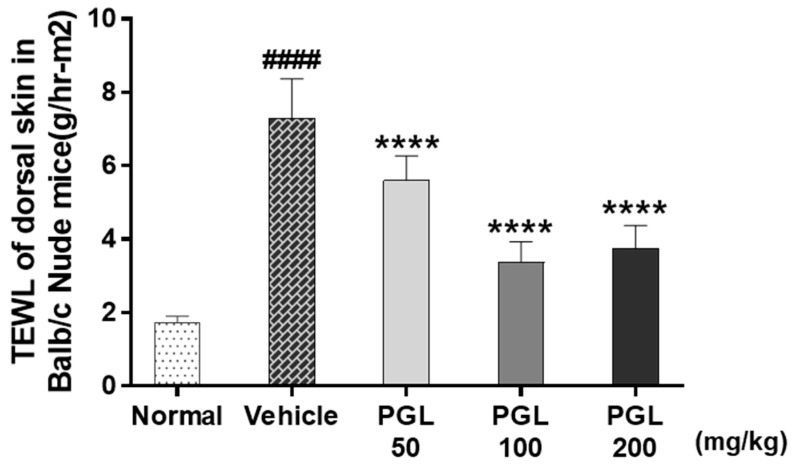
The effect of PGL on epidermal TEWL by TM 300 Tewameter. Values are mean ± S.E.M. ^####^
*p* < 0.001 vs. normal group. **** *p* < 0.001 vs. vehicle group.

**Figure 6 molecules-28-06734-f006:**
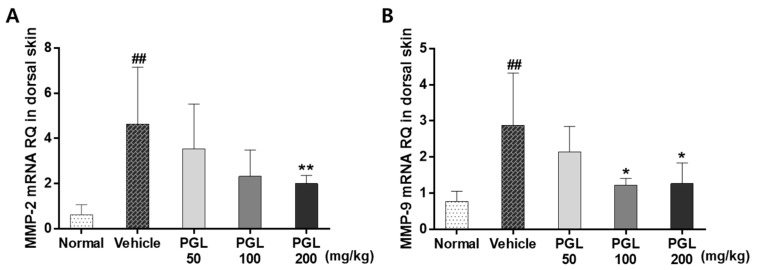
Effects of PGL on matrix metalloproteinase-2 (**A**) and -9 (**B**) mRNA expression in the skin of mice exposed to UVB. Values are expressed as the mean ± S.E.M. for five mice. ^##^
*p* < 0.001 vs. normal group. * *p* < 0.05 vs. vehicle group. ** *p* < 0.01 vs. vehicle group.

**Figure 7 molecules-28-06734-f007:**
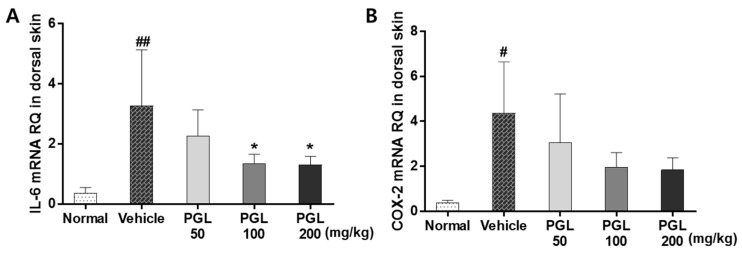
Effects of PGL on IL-6 (**A**) and cyclooxygenase-2 (**B**) mRNA expression in the skin of mice exposed to UVB. Values are expressed as the mean ± S.E.M. for five mice. ^#^
*p* < 0.05 vs. normal group. ^##^
*p* < 0.001 vs. normal group. * *p* < 0.05 vs. vehicle group.

**Table 1 molecules-28-06734-t001:** The quantity of ginsenosides of processed ginseng leaf extract and ginseng leaf extract.

	Content (mg/g)	
	Processed Ginseng Leaf Extract	Ginseng Leaf Extract
Rg1	10.5 ± 3.30	74.2 ± 2.79
Re	30.00 ± 0.85	167.7 ± 1.24
Rg2	39.3 ± 1.77	32.4 ± 0.98
Rb1	4.9 ± 0.06	15.6 ± 0.17
Rc	11.30 ± 0.01	37.2 ± 0.21
Rb2	23.5 ± 1.98	54.0 ± 2.07
Rb3	5.0 ± 0.11	12.6 ± 0.16
Rd	83.7 ± 2.31	125.1 ± 4.19
Rg3	29.4 ± 1.50	2.1 ± 0.00
Rk1	35.2 ± 1.76	-

**Table 2 molecules-28-06734-t002:** Sequences of the primers used for the real-time polymerase chain reaction.

Gene		Primer Sequence
MMP-2	Forward	5′-CAG GGA ATG AGT ACT GGG TCT ATT-3′
	Reverse	5′-ACT CCA GTT AAA GGC AGC ATC TAC-3′
MMP-9	Forward	5′-AAT CTC TTC TAG AGA CTG GGA AGG AG-3′
	Reverse	5′-AGC TGA TTG ACT AAA GTA GCT GGA-3′
IL-6	FAM	5′-CTGTGTAATGAAAGACGGCACACCCACC-3′
COX-2	Forward	5′-ATG GAT CGA AGA CTA CGT GCA A-3′
	Reverse	5′-GGG ATT TCC CAT AAG TCC TTT C-3′

## Data Availability

The data presented in this study are available in this article.

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
