# Peer review of "Photoprotective Effects of Processed Ginseng Leaf Administration against UVB-Induced Skin Damage in Hairless Mice"

_molecules, 2023, doi:10.3390/molecules28186734_

Round 1

Reviewer 1 Report

I have several comments before accepting the authors' submitted study. Please carefully check and respond to them.

-Comment 1. Language correction has to be done extensively.

-Comment 2. The abstract is unclearly written. I think you should check carefully the using of expressions. Like you have used beneficial effects on human. Which functions related to your manuscript. Also, oral administration is a wide range of application. You should be precise to your study. Five lines for the introduction of abstract is too much. Which methods you used should be clarified in the abstract in brief. In the conclusion section of abstract the authors did not mention that the study provided scientific guidance for comprehensive applications of the Oral administration of ginseng leaves. Could you clearly mention how it could be with more attractive example, not just proteinase attenuation? You have used one variety and that could not be obtained with that number of varieties. Please focus on your study novel findings and objectives, not in the general term expressions like proteinase attenuation and anti-inflammation.     

-Comment 3. Introduction needs more clarification for the importance of the used ginseng geographical varieties. Please define in detail the plant, its utility and the research in the modern context with proper references. Also, the issue in the introduction, why you focused on the biomarkers like MMP-2 and MMP-9 Secretion. You should also mention the functional compounds and bioactives application in the introduction section.

-Comment 4. What was the clear hypothesis of this study? Why it was being studied?

-Comment 5. Was the Ginseng leaves variety have been collected and prepared by the authors at the same time? Why you did not mention the general characteristics of your variety. Your sample study should be defined in more details with photos of the used leaves in flow chart figure. 

-Comment 6. Table 1, please add the SD. How many replicates have you did for this experiment? Figure 3 is not well organized. You should clarify the labels. Normal and PGL 200 colors should be more differentiated. Why do you think that 16W (****p < 0.001) PGL 200 increased again after the significant decrease of PGL 100.

-Comment 7. More details about the reason to recommend protective impact in lines 106-117.

-Comment 8. The figures are not with good quality.

-Comment 9. The interpretation of figure 2 is unclear and not sufficient.

-Comment 10. The authors should provide a flowchart or schematic diagram of the mode of action.

-Comment 11. The discussion section needs more work for bioactivities, authors have mainly just described the results.

 -Comment 1. Language correction has to be done extensively.

Author Response

I revised the manual according to your comment, and the contents of the correction are described in the attachment.

Reviewer 2 Report

Dear Nithinan Sawettanai,

The manuscript by Eunjung Son  et al. entitled “Photoprotective effects of processed ginseng leaf administra-tion against UVB-induced skin damage in hairless mice”, aimed to demonstrate the effectiveness of ginseng compound against wrinkle 21 depth, epidermal thickness, and trans epidermal water loss levels.

It is reviewer opinion that the paper is well written and that the authors demonstrated the effect of Rg3 and Rk1 on HaCat cell lines and on mice skin. Only few correction need to be done in my opinion:

- figure 1: it is possible to enlarge the graph?

- figure 3, 4, 5, 6, 7: please add the unit of measurement in corrspondance of PGL concentrations;

- paragraph 3.2: can the authors  to specify the difference between processed ginseng leaf extract and Ginseng leaf extract?

- discussion: can the authors improve the discussion of the obtained results increasing the bibliography?

It is reviewer opinion that the paper can be accepted.

Best regards

Author Response

(The authors gave the same response as above.)

Round 2

Reviewer 1 Report

 -

 -

Author Response

According to your comment,  the chromatogram was added as supplement.
